# Validation of TaqMan-Based Assays for Specific Detection and Differentiation of Wild-Type and Neethling Vaccine Strains of LSDV

**DOI:** 10.3390/microorganisms9061234

**Published:** 2021-06-06

**Authors:** Dejan Vidanović, Bojana Tešović, Milanko Šekler, Zoran Debeljak, Nikola Vasković, Kazimir Matović, Andrey Koltsov, Kiril Krstevski, Tamaš Petrović, Ilse De Leeuw, Andy Haegeman

**Affiliations:** 1Veterinary Specialized Institute Kraljevo, 36000 Kraljevo, Serbia; tesovic@vsikv.com (B.T.); milankosekler@yahoo.com (M.Š.); debeljak@vsikv.com (Z.D.); vaskovic@vsikv.com (N.V.); matovic@vsikv.com (K.M.); 2Federal Research Center of Virology and Microbiology, 601125 Pokrov, Russia; kolcov.andrew@gmail.com; 3Faculty of Veterinary Medicine, University Ss Cyril and Methodius in Skopje, 1000 Skopje, North Macedonia; krstevski@fvm.ukim.edu.mk; 4Scientific Veterinary Institute Novi Sad, 21000 Novi Sad, Serbia; tomy@niv.ns.ac.rs; 5Sciensano, 1050 Brussels, Belgium; ilse.deleeuw@sciensano.be (I.D.L.); andy.haegeman@sciensano.be (A.H.)

**Keywords:** lumpy skin disease virus, real-time PCR, DIVA diagnostic protocol, differentiation, field strain, vaccine strain

## Abstract

Lumpy skin disease (LSD) is an important animal disease with significant health and economic impacts. It is considered a notifiable disease by the OIE. Attenuated strains of LSDV have been successfully used as vaccines (LAV) but can also produce mild or systemic reactions. Vaccination campaigns using LAVs are therefore only viable if accompanying DIVA assays are available. Two DIVA qPCR assays able to distinguish Neethling-based LAVs and wild-type LSDV were developed. Upon validation, both assays were shown to have high sensitivity and specificity with a diagnostic performance comparable to other published DIVA assays. This confirmed their potential as reliable tools to confirm infection in animals during vaccination campaigns based on Neethling vaccine strains.

## 1. Introduction

Due to its significant health and economic impacts, lumpy skin disease (LSD) is considered an important animal disease that is on the OIE’s list of notifiable diseases. The etiological agent of the disease is lumpy skin disease virus (LSDV), which belongs to the family of Poxviridae [1] and is highly host-specific, causing disease in cattle and water buffalo. Together with sheeppox virus (SPPV) and goatpox virus (GTPV), it belongs to the genus of *Capripoxvirus*. These viruses have a DNA genome about 150,000 bp long that encodes 147 putative genes [2].

Lumpy skin disease was first described in Zambia in 1929 and exclusively occurred on the African continent and the Middle East until 2012, when it began to spread to Turkey (2013); Iraq and Cyprus (2014); Greece (2015); Bulgaria, N. Macedonia, Serbia, Albania, Montenegro, Armenia, Azerbaijan, Kazakhstan, and Russia (2016); Namibia, S. Arabia, and Mozambique (2017); Bangladesh, China, India, and Syria (2019); Bhutan, Nepal, Djibouti, Vietnam, Hong Kong, Myanmar, and Sri Lanka, (2020) [3,4,5]. The disease is mainly characterized by the development of nodules on the skin that can develop into skin lesions and scab formation. Mortality is generally low, but exceptions, as well as differences between *Bos taurus* and *Bos indicus* breeds, have been reported [1,6,7,8,9,10,11].

In LSD control, attenuated strains of LSDV, SPPV, and GTPV have been successfully used as vaccines in infected areas [1]. When using a vaccine-based on the Neethling strain, mild or systemic reactions have been observed in some vaccinated animals. These reactions can cause complications with control and eradication measures, such as stamping out, as the distinction between vaccinated and infected animals based on clinical picture becomes problematic [5,12]. These issues can be addressed by the application of diagnostic assays that quickly and specifically differentiate between wild-type LSDV strains and vaccine LSDV strains.

Molecular assays such as PCR have the advantages of being quick, being highly sensitive, and having the ability to target highly specific genomic regions. Though the genomes of capripoxviruses are highly conserved [2], the increasing availability of (full genome) sequences allow for the identification of genomic differences between wild-type LSDV and vaccine strains. These differences can be used for the development of PCRs that can differentiate infected from vaccinated animals (DIVA). Several such DIVA PCRs have been developed, including conventional [13] and real time PCRs using specific TaqMan probes [14,15,16,17]. An overview of all these tests was described in detail elsewhere [18]. However, the cost and/or the readily availability of the primers, probes, and reagents can be a bottleneck, especially if they make use of modified nucleotides such as LNAs [15]. Therefore, it was the aim of this study to develop and validate an assay with standard real-time PCR primers, probes, and reagents suited for the rapid, sensitive, and specific detection and differentiation of wild-type LSDV strains from Neethling-based vaccine strains of LSDV. The KV-DIVA qPCR protocol consisted of two TaqMan probe-based assays. The first assay was specific for wild-type strains (KV-2 assay) and was a modified version of a published assay [14]. The second assay was specific for Neethling vaccine strains (KV-VAC), was developed in 2017, and has not yet been published. Both assays were not developed as first line diagnostic tools but rather as DIVA tests in combination with assays specific for all Capripox viruses [19,20,21,22] to support control and eradication strategies. Another import aspect of an assay is its robustness. The assay will be implemented in different laboratories, each with their own peculiarities and equipment. This can cause minute changes in test circumstances and can potentially influence assay characteristics. This was evaluated during this study by performing the validation in two separate and independent laboratories, namely in VSI “Kraljevo” (Serbia) and Sciensano (Belgium).

## 2. Materials and Methods

### 2.1. Design of KV-2 and KV-VAC Assays

The design of primers and tests for both tests was performed in VSI “Kraljevo” using the online software Primer3 (primer3.ut.ee) and Primer Blast (https://www.ncbi.nlm.nih.gov/tools/primer-blast/, accessed on 1 March 2021).

The design of primers and probes for wild-type strain-specific assay, KV-2, was previously described [14]. After the sequencing of twenty wild-type LSDV strains obtained from IAEA laboratory in Seibersdorf, Austria, (data not shown), forward and reverse primers were slightly modified compared to the original protocol. Modifications are marked in bold letters and presented in Table 1.

Primers and probes specific for the vaccine strains of LSDV (KV-VAC assay) were designed to detect part of the LW008 gene, where the probe (labeled with FAM) binds 100% specifically for part of the genome of vaccine strains; see Table 2. In wild-type strains of LSDV, as well as in SPPV and GTPV strains at the same position in the genome, there are 9 nucleotides mismatches, and the probe does not bind to LW008 gene of the mentioned viruses.

### 2.2. Validation Study 

The validation study was performed in parallel at the Veterinary Specialized Institute Kraljevo (VSI Kraljevo) and Sciensano (EU Reference laboratory for Capripox viruses).

#### 2.2.1. DNA Extraction

The extraction of viral DNA from blood samples, milk, skin scarifications, skin biopsies, internal organs, and nasal swabs of animal origin, as well as swabs and samples from the environment and insects, was performed at VSI by using a commercial MagVet Universal DNA/RNA kit (LSI, Germany) and Bioextract Superball (Biosellal, Dardilly, France) on Kingfisher mL and Kingfisher Flex (Thermo Fisher Scientific, Vantaa, Finland) according to the manufacturer’s instructions. The extracted DNA was stored at temperatures below −16 °C until the start of the testing.

Viral DNA was extracted at Sciensano using the NucleoSpin Blood and NucleoSpin tissue kits (Macherey-Nagel, Düren, Germany) as previously described [21].

#### 2.2.2. Samples for Analytical Sensitivity (Ase), Specificity (Asp), Efficiency, and Repeatability

The efficiency of the qPCR test performed at VSI for wild-type strains of LSD was performed by examining serial tenfold dilutions of wild-type strain of virus Serbia/Bujanovac/2016 [23]. The efficacy of the qPCR test for vaccine strains of LSD was performed by testing serial ten-fold dilutions of a vaccine based on the Neethling strain of LSD (OBP, South Africa, NCBI Accession No. AF409138). The obtained results were processed in the AriaMx software.

The analytical sensitivity of the assays was determined at VSI Kraljevo by testing serial dilutions of quantified wild-type and vaccine strain DNA obtained from the EU reference laboratory for Capripox viruses, Sciensano, Belgium. The number of copies of the LSDV Bulgaria wild-type strain was 1.15 × 10^6^/µL, and the number of copies of the genome of the LSDV Neethling vaccine strain was 5.63 × 10^6^/µL. The cut-offs of the assays were determined by testing of 20 replicates of wild-type strain Serbia/Bujanovac/2016 and Neethling vaccine strain (OBP vaccine) diluted to a limit of analytical sensitivity and calculating 2 SD from the mean Ct values.

The repeatability was determined at VSI by analyzing eight replications in triplicate. The used samples were the OBP vaccine strain (KV-VAC) and the Serbia/Bujanovac/2016 LSDv strain (KV-2). At Sciensano, the repeatability and reproducibility were evaluated by spiking negative EDTA-blood with a wild-type LSDV strain (LSDV Bulgaria) and a Neethling vaccine strain. From these 2 samples, genomic DNA was extracted, and the 2 DIVA qPCRs (KV-2 and KV-VAC) were run in 4-fold (repetitions, equal conditions) in 5 different runs (with different reproduction conditions).

The analytical specificity of the assays was tested on wild-type strains of LSDV from Serbia, as well as on different wild-type LSDV strains obtained from IAEA laboratory in Seibersdorf, Austria, and Diagnostic Veterinary Laboratory, Podgorica, Montenegro. The specificity was also checked at the Federal Research Center of Virology and Microbiology, Pokrov, Russia on their collection of LSDV, SPPV, GTPV strains, and other viruses, as well as at the Faculty of Veterinary Medicine, Ss. Cyril and Methodius University, Skopje, North Macedonia. Sciensano tested their own collections, as well as samples provided in 2004 by Dr. P. Kitching, who was then working at the National Centre for Foreign Diseases, Winnipeg, Manitoba, Canada; in collaboration with Dr. P. Mellor, then working at the Pirbright Institute (Pirbright, United Kingdom); and in 2010 by Dr H. Yadin, then working at the Kimron Veterinary Research Institute (Bet Dagan, Israel). These viruses were isolated over the past 4 decades from sheep, goat, and cattle throughout the endemic regions of Africa, Asia, and the Middle East. In total, 90 Capripox virus strains were tested (Appendix A) In addition, a panel of 41 other microorganisms (Appendix A) were tested with the KV-2 and KV-VAC real-time PCRs.

#### 2.2.3. Samples for Diagnostic Sensitivity (Dse) and Specificity (Dsp)

The diagnostic sensitivities of the KV-2 and KV-VAC were determined at VSI by examining samples originating from naturally infected animals (blood, nasal swabs, skin biopsy, and milk), vaccinated animals, insects that may be carriers of this disease, and environmental swabs. The number and type of samples used for test validation are given in Table 3.

The diagnostic sensitivity was tested at Sciensano using 48 samples (tissue and blood) obtained from animal experiments with LSDV (Appendix A). The panel consisted of 18 wild-type and 30 vaccine-type samples. Though the complete range of very strong to very weak positive samples were included, emphasis (regarding number of samples) was placed on weak to very weak positives because these are the most challenging for a PCR.

For the diagnostic specificity, 34 negative blood samples originating from Belgian cattle were tested at Sciensano.

#### 2.2.4. Samples for Comparative Analysis

The diagnostic sensitivity and specificity were calculated at VSI by comparing the results of the KV-2 and KV-VAC assays with results obtained with the CaPV Bowden assay on 163 samples listed in Appendix A. At Sciensano, the panels for diagnostic sensitivity were used with the KV-2 and KV-VAC qPCR, the DIVA Agianniotaki PCR [12], and the panD5R PCR [21]; see Appendix A

### 2.3. Real-Time PCR

#### 2.3.1. KV-2 and KV-VAC Assays

The KV-2 and KV-VAC assays were evaluated as single-plex in both institutes. The PCR reaction mix at VSI contained 12.5 µL of Brilliant III Ultrafast qPCR master mix (Agilent Technologies, West Cedar Creek, TX, USA), 0.2 µL of the forward (50 µM) and reverse primer (50 µM), and 0.1 µL of the probe (50 µM). Water was added to a total volume is 20 µL, and 5 µL of template were added. The real-time PCR reaction was performed on AriaMx (Agilent Technologies, Singapore) and Stratagene Mx3000P (Agilent Technologies, Waldbronn, Germany) devices. Primers and probes were ordered from Metabion, Germany. The thermal profile for both assays was hot start for 3 min at 95 °C followed by 45 cycles of denaturation for 15 s at 95 °C and annealing for 30 s at 60 °C.

Real-time PCRs at Sciensano were carried out in a total volume of 20 µL, consisting of 10.0 µL of LightCycler 480 (LC480) Probes Master (Roche, Vilvoorde, Belgium), 2.5 µL of DNA template, 1 U FastStartTaq DNA polymerase (Roche, Vilvoorde, Belgium), 0.8 mM MgCl2, a final primer concentration of 1 µM and a final probe concentration of 0.35 µM. BSA was added additionally to a final concentration of 0.1 µg/µL. The primers and DNA template were separately denatured at 95 °C for 3 min before the rest of the mix was added. The thermal cycling profile for both real-time PCRs was: 95 °C for 10 min followed 45 cycles of 95 °C for 15 s and 60 °C for 30 s. The real-time PCRs were carried out on the Roche LightCycler 480 instrument. Primers and probes were ordered from IDT, Belgium.

#### 2.3.2. Additional Real-Time PCRs

At VSI, a published panCapripox real-time PCR Bowden assay was used to confirm the presence of Capripox virus [19].

At Sciensano, a panCapripox-specific D5R assay [21] was used to demonstrate Capripox DNA. The DIVA real-time PCR of Agianniotaki [15] was used to compare the differentiation capacity of the KV-2 and KV-VAC assays.

## 3. Results

### 3.1. Results of Validation Study in VSI Kraljevo

#### 3.1.1. Efficiency of the Wild-Type KV-2 Strain Assay

By testing the five tenfold serial dilutions of the Serbia/Bujanovac/2016 isolate using AriaMx software, it was determined that the efficiency of the qPCR reaction, using Brilliant III Ultrafast qPCR kit, was 99%.

#### 3.1.2. Efficiency of the Vaccine KV-VAC Strain Assay

By testing the five tenfold serial dilutions of the OBP vaccine strain of the LSD virus (Neethling strain), using AriaMx software, it was determined that the efficiency of the qPCR reaction, using the Brilliant III Ultrafast qPCR kit, was 99.6%.

#### 3.1.3. Analytical Sensitivity of the Assays

An analysis of the obtained results for 95% probability of detection (20 replicates) determined that the limit of detection of the test for the wild-type LSDV strains KV-2 was 24 copies of the genome per reaction, while the limit of detection of the assay for the KV-VAC strains was 12 copies of the genome per reaction.

#### 3.1.4. Determination of Cut-off of Assays

Based on the testing of 20 replicates of positive quantified reference DNA (LSDV Bulgaria as wild-type strain and Neethling strain as vaccine strain) diluted to the limit of analytical sensitivity of both tests (24 copies of genome/reaction for the wild-type strain and 12 copies of the genome/reaction for vaccine strains of LSDV), cut-off values were determined for both assays to be Ct (Cq) 40. This meant that any sample that had a Ct (Cq) value of 40 or lower was considered positive.

#### 3.1.5. Determination of Repeatability and Reproducibility of the Assays

The wild-type and vaccine LSDV strains were correctly identified 20 times with the real-time PCR assays. No cross contamination was found between the wild-type and vaccine strains. The evaluation of the repeatability of the KV-2 and KV-VAC assays is summarized in Table 4 and Table 5. A high reproducibility was found for both real-time PCRs. The inter-run coefficients of variation for the KV-2- and KV-VAC-type assays were 0.79% and 0.47%, respectively.

#### 3.1.6. Diagnostic Sensitivity and Diagnostic Specificity

##### Diagnostic Sensitivity

The results of the comparative examination of samples originating from diseased animals, environmental samples, and insects using the Bowden, KV-2, and KV-VAC assays are given in Appendix A. A total of 163 samples, including naturally infected animals, vaccinated animals, flies, and environmental swabs taken during 2016 outbreak, were analyzed. When using the Capripox-specific Bowden assay, 149 samples scored positive and 14 scored negative. When these 163 samples were analyzed with both new assays, 115 (KV-2 assay) and 32 (KV-VAC assay) samples were found to be positive. All samples that were negative for the Bowden assay were also negative for both assays. Three samples (one blood sample and two environmental swabs) that were positive for the Bowden assay, albeit with a Ct value of 36.5 or higher, were negative for both assays. As these samples originated from a non-vaccinated animal (blood sample) or farms where vaccination was not performed (swabs), a signal was expected in the KV-2 assay. Three samples, two skin samples and one blood sample, were positive for both the wild-type and vaccine strains of LSDV.

Using the PCR data of these 163 samples, the diagnostic sensitivity and specificity of the KV-2 and KV-VAC assays were compared to the Bowden assay using MedCalc statistical software. The diagnostic sensitivity of the KV-2 assay was 97.46%, with a diagnostic specificity of 100%, while the diagnostic sensitivity and specificity of the KV-VAC assay were both 100%. The results are summarized in Table 6.

An additional 40 LSDV-negative samples and 40 samples from vaccinated cattle were tested with the Bowden, KV-2, and KV-VAC assays. The specificity and sensitivity of the KV-2 and KV-VAC assays compared with the Bowden assay was 100%. Results are given in Table 7.

#### 3.1.7. Analytical Sensitivity and Exclusivity

The KV-2 assay, which is specific only for wild-type strains of LSDV, successfully detected all 31 wild-type strains of LSDV and did not detect any of the three Neethling vaccine strains, 12 SPPV strains, and 12 GTPV strains (Appendix A). However, the Kenyavac KSGP 0240 strain, which is used as a vaccine in some African countries, scored positive for the KV-2 assay. Additionally, the KV-2 assay did not detect the recombinant Dergachevskyi strain, the only recombinant strain used in this study.

The LSDV KV-VAC assay specific for vaccine strains detected all three Neethling virus vaccine strains used in this study, but it did not detect Kenyavac KSGP 0240 as a vaccine strain. This assay did not detect any of the 31 wild-type strains of LSDV, and it did not detect any of the 12 SPPV strains and 12 GTPV strains. Additionally, the KV-VAC assay did not detect the Dergachevskyi recombinant strain.

The additional exclusivity testing of the KV-2 and KV-VAC assays with 20 bacterial and 10 viral species (Appendix A) showed no false-positive results.

#### 3.1.8. Statistical Analysis of VSI Kraljevo Validation Results

Passing–Bablok regression analysis revealed the following equations between the Ct values obtained by the BOW real-time PCR (y), the KV-2 assay (x_1_) (Figure 1a), and the KV-VAC assay (x_2_) (Figure 2a) with 95% CI:(1)y=−0.01 (95% CI:−0.36−0.39)+1.06 (95% CI: 1.05−1.08) x1
(2)y=−0.38 (95% CI:−1.11−0.20)+1.02 (95% CI: 1.00−1.04) x2

Respective Bland–Altman plots are presented in Figure 1b and Figure 2b. The average total bias for the comparison of the BOW real-time PCR with the KV-2 assay was −1.63 cycles (Figure 1b), and that with the KV-VAC assay was −0.05 cycles (Figure 2b). According to concordance correlations, the precision between methods, which measured how far each observation deviated from the best-fit line and represent Pearson correlation coefficients, were 97.05% and 99.68% when the BOW real-time PCR was compared with the KV-2 assay and with KV-VAC assay, respectively.

The average Ct values obtained by the BOW real-time PCR were not significantly different from those obtained with the VAC real-time PCR (mean ± SE: 27.17 ± 0.95 and 27.22 ± 0.96 for the BOW and VAC assays, respectively; *p* > 0.05). However, the average Ct values obtained with the BOW assay were slightly lower than those obtained with the KV2 method and this difference was statistically significant (mean ± SE: 25.39 ± 0.66 and 27.01 ± 0.71 for BOW and KV-2, respectively; *p* < 0.05).

### 3.2. Results of Validation Study in Sciensano

#### 3.2.1. Determination of Repeatability and Reproducibility of the Assays

The wild-type and vaccine LSDV strains were correctly identified 20 times with the real-time PCRs. No cross contamination was found between wild-type and vaccine strains. The total coefficient of variation was 2.16% for the wild-type real-time PCR and 1.98% for the vaccine-type real-time PCR. A high reproducibility was found for both real-time PCRs. The results are shown in Table 8 and Table 9.

#### 3.2.2. Diagnostic Sensitivity and Diagnostic Specificity

##### Diagnostic Sensitivity

Forty-eight samples with a positive status for the panDR real-time PCR were analyzed. The KV-VAC real-time PCR correctly detected 25 out of the 30 vaccine samples and all wild-type samples (*n* = 18). The five vaccine samples that were negative for the KV-VAC real-time PCR had a very low viral load, as indicated by the panD5R real-time PCR (Cp-values > 38). The KV-2 real-time PCR detected all the wild-type samples (*n* = 18) and all vaccine samples (*n* = 30) correctly.

The DIVA Agianniotaki real-time PCR correctly detected 20 out of the 30 vaccine samples and all wild-type (*n* = 18) samples. The 10 vaccine-type samples that were negative for the vaccine channel had a very low viral load, as indicated by the panD5R real-time PCR (Cp-values > 38). All obtained PCR results are summarized in Appendix A

##### Diagnostic Specificity

All blood samples scored negative in the KV-2 and KV-VAC real-time PCRs, except one sample that had a borderline Cp value of 40 in the KV-2 real-time PCR. For DIVA Greece, one sample had a Cp value of 40 for the wild-type channel, and another sample had the same value for the vaccine channel. For the panD5R real-time PCR, one sample was doubtful in the D5R channel. All the obtained values for internal and external control with the panD5R real-time PCR were within the acceptance criteria. The obtained Cp values of 40 for the different real-time PCRs could indicate a non-specific reaction.

The diagnostic sensitivity and specificity of the KV-2 and KV-VAC assays compared to the D5R assay were calculated using the MedCalc statistical software, and the results are given in Table 10. The diagnostic sensitivity of the KV-2 assay was 100%, while the diagnostic sensitivity of the KV-VAC assay was 85.71%, with a specificity of 100% for both.

The diagnostic sensitivity and diagnostic specificity of the DIVA Agianniotaki assay compared to the D5R assay were calculated using the MedCalc statistical software, and the results are given in Table 11. The diagnostic sensitivity of the wild-type Agianniotaki assay was 100%, while the diagnostic sensitivity of the Agianniotaki vaccine assay was 75%, with a specificity of 100% for both.

#### 3.2.3. Analytical Sensitivity and Exclusivity

None (*n* = 34) of the SGPVs, SPPVs, and GPVs were detected with the KV-2 and KV-VAC real-time PCRs, while they were clearly positive with the pan-Capripox D5R (Appendix A).

##### Exclusivity of the Assays

None of the samples (*n* = 11) from animals infected with clinically or genetically related pathogens were detected with the KV-2 and KV-VAC real-time PCRs (Appendix A).

#### 3.2.4. Statistical Analysis of Sciensano Validation Results

Passing–Bablok regression analysis (Figure 3a, Figure 4a, Figure 5a and Figure 6a) revealed the following equations between the Ct values obtained by the D5R real-time PCR (*y*), wild-type KV-2 (*x*_1_), wild-type Agianniotaki (*x*_2_), KV-VAC (*x*_3_), and Agianniotaki vaccine (*x*_4_) assays with 95% CI:(3)y=−1.68 (95% CI:−29.54−17.81)+1.03 (95% CI: 0.48−1.81) x1 
(4)y=−14.38 (95% CI:−86.57−8.26)+1.39 (95% CI: 0.76−3.37) x2 
(5)y=4.08 (95% CI:−1.62−9.03)+0.89 (95% CI: 0.76−1.05) x3 
(6)y=5.36 (95% CI:−3.10−16.62)+0.88 (95% CI: 0.57−1.13) x4 

The respective Bland–Altman plots are presented in Figure 3b, Figure 4b, Figure 5b and Figure 6b. The average total biases were 0.8, −0.2, −0.2, and −0.9 cycles for the comparisons of the D5R real-time PCR with wild-type KV-2, D5R real-time PCR with wild-type Agianniotaki, D5R real-time PCR with KV-VAC, and D5R real-time PCR with Agianniotaki vaccine, respectively. According to concordance correlation coefficient, the precision between methods, which measures how far each observation deviates from the best-fit line and represents Pearson correlation coefficient, was 58.70%, 48.83%, 93.98%, and 82.35% when the D5R real time PCR was compared with the wild-type KV-2, wild-type Agianniotaki, KV-VAC, and with Agianniotaki vaccine, respectively.

The average Ct values obtained by the D5R real-time PCR were not significantly different from those obtained with the real-time PCR of wild-type KV-2 (mean ± SE: 35.96 ± 0.49 and 35.17 ± 0.5652 for the D5R and wild-type KV-2 assays, respectively; *p* > 0.05). The average Ct values obtained by D5R real-time PCR were not significantly different from those obtained with the real-time PCR of wild-type Agianniotaki (mean ± SE: 35.96 ± 0.49 and 36.17 ± 0.67 for the D5R and wild-type Agianniotaki assays, respectively; *p* > 0.05). The average Ct values obtained by D5R real-time PCR were not significantly different from those obtained with the real-time PCR of KV-VAC (mean ± SE: 36.88 ± 0.75 and 37.05 ± 0.68 for the D5R and KV-VAC assays, respectively; *p* > 0.05). The average Ct values obtained by D5R real-time PCR were not significantly different from those obtained with the real-time PCR of Agianniotaki vaccine (mean ± SE: 36.08 ± 0.84 and 36.95 ± 0.74 for the D5R and Agianniotaki vaccine assays, respectively; *p* > 0.05).

## 4. Discussion

Lumpy skin disease is an important disease of cattle and a major threat to livestock with severe socio-economic impacts. The direct and indirect costs of disease control can cause great economic damage. Experience from the recent outbreak in the Balkan region shows that the mass and broad vaccination of animals is an efficient way to eradicate this disease [5,24]. The rapid detection and reliable differentiation of wild-type from vaccine strains comprise one of the key factors for the application of timely measures to combat LSD [14]. There is a need for diagnostic tests to be constantly improved in order to be able to respond to the ever-changing situation in the field. During the eradication of LSD in the epidemic of Serbia in 2016, vaccination began three weeks after the first confirmed case of the disease. A coverage of 99% was achieved three months after the start of the campaign, when the last case was recorded [5,24]. The average time between the notification of a suspicion to stamping out was 2.2 days, which, with the application of mass vaccination, had a decisive influence on the rapid suppression of the disease [5]. The reason for such a rapid diagnosis was the development of two TaqMan-based qPCR tests for the specific detection of wild-type strains of LSDV [14], which allowed for results to be obtained in a few hours. This methodology allows for a short test duration and a high throughput of samples.

The assays presented in this paper are not intended for first-line screening of samples but are intended to be used after examination of samples by another Pan-Capripox essay, such as [19,20,21,22] with usage of internal amplification control (IAC) to exclude influence of PCR inhibition. KV-DIVA assays were validated in the Veterinary Specialized Institute Kraljevo, NRL for Capripox viruses in Serbia and in Sciensano, European Union Reference Laboratory for Capripox viruses in Belgium.

In VSI Kraljevo, validation was performed on 243 samples of blood, skin, nasal swabs, and milk originating from naturally infected or vaccinated animals, insects, and environmental swabs. Tests have shown a very high sensitivity. The KV-2 test for wild-type strains showed a sensitivity of 24 copies of the LSDV genome per reaction, while the KV-VAC test, specific for vaccine strains, showed a sensitivity of 12 copies of the LSDV genome, which was comparable to the results obtained by [15], who reported the sensitivity of the DIVA test of eight genome copies of the wild-type or vaccine LSDV strain per reaction. Pestova and Sprygin [16,17] reported limits of detection of 0.21 and 0.15 lg TCD50/mL for wild-type and vaccine assays, respectively. The developed KV-VAC assay showed a very good linear correlation with the Bowden assay, and the obtained Ct values were almost identical, which was expected because the limit of detection of both assays was the same. The wild-type strain-specific KV-2 assay was somewhat less sensitive than the Bowden assay (24 vs. 12 copies, respectively), and therefore it is understandable why there were statistically significant differences in Ct values. The observed differences are more noticeable in samples with lower quantities of the viral genome, especially in those with a Ct value above 33 with the Bowden assay.

In Sciensano, validation was performed on 48 samples obtained from animal experiments by comparing the performance of the KV-2 and KV-VAC assays with the DIVA Agianniotaki and CaPV D5R assays. When comparing with performance of wild-type LSDV strains with the D5R assay, the KV-2 and DIVA Agianniotaki assays showed 100% sensitivity. When comparing the performance of the LSDV vaccine strains with that of the D5R assay, the KV-VAC and DIVA Agianniotaki assays showed 85.71% and 75% sensitivity values, respectively. After comparing KV-DIVA assays with DIVA Agianniotaki assay, it could be concluded that the KV-DIVA assays had almost identical performances. It must be emphasized that mainly samples with high Ct values were selected for the experiment. There were no statistically significant differences between Ct values obtained with panCapripox D5R assay and the KV-DIVA or Agianniotaki DIVA assay.

The specificity of the tests in VSI Kraljevo was examined on 60 different Capripox viruses, 31 of which belonged to classical wild-type strains of LSDV, four of which belonged to vaccine strains of LSDV (three based on Neethling and one based on the KSGP 0240 strain), one of which was a recombinant strain of LSDV, 12 of which were SPPV strains, and 12 of which were GTPV strains. All classical wild-type strains of LSDV and all Neethling-based vaccine strains were accurately identified using the KV-2 and KV-VAC tests.

Using the KV-DIVA assays, the Kenyavac KSGP 0240 strain was identified as a wild-type strain. Though this strain is used as a vaccine against LSD in Kenya and some other African countries, genome studies have shown that this strain is significantly genetically different from the Neethling-based vaccine strains of LSDV. Scientific and clinical trials have led to the conclusion that this strain is not sufficiently attenuated and causes clinical symptoms of disease in vaccinated animals [25].

The specificity of the tests in Sciensano was examined on 34 CaPV strains, six of which were wild-type strains of LSDV, four of which were vaccine strains of LSDV, 20 of which were SPPV strains, and four of which were GTPV strains. All wild-type and vaccine strains of LSDV were accurately identified, while there were no false-positive results.

KV-2 and KV-VAC assays were developed in 2016 and 2017 when possible recombinant strains had not yet been reported and aimed to correctly distinguish disease-causing wild-type strains from vaccine Neethling strains of LSD virus. These assays failed to detect the Dergachevsky recombinant strain, the only recombinant strain available to us for in-vitro testing. Byadovskaya [26] conducted an extensive comparative performance test of several commercial and published DIVA tests, and they found that they were unable to correctly identify recombinant strains. Badhy [27] stated that the isolates that appeared in 2019 in Bangladesh differed from strains from Africa, Europe, and the Middle East, as well as from variant strains from Russia and China [28,29]. The occurrence and spread of recombinant strains have not been fully elucidated, and their circulation must be monitored.

In the most European countries where the disease appeared in 2015 and 2016, a live attenuated vaccine based on the Neethling strain was used. Therefore, it was very important to use tests in diagnostic laboratories to distinguish wild-type LSDV from Neethling vaccine strains. In this paper, we presented an extensive validation study that included 90 different Capripox virus strains and 291 samples originating from naturally infected animals, experimentally infected animals, vaccinated animals, insects, and environment samples, conducted in two laboratories. The validation in the two laboratories with different equipment demonstrated not only the high sensitivity and specificity of the presented DIVA tests but also their robustness. This guarantees their accuracy and cost-effectiveness (essential for disease eradication), and it demonstrates their flexibility in application under changing laboratories conditions. These DIVA assays are also in routine use for strain typing at the Federal Research Center of Virology and Microbiology, Pokrov, Russia, the Faculty of Veterinary Medicine, Ss. Cyril, and Methodius University, Skopje, North Macedonia. However, in countries using an LSD vaccine based on SpPV or GtPV, a different diagnostic strategy must be applied.

## Figures and Tables

**Figure 1 microorganisms-09-01234-f001:**
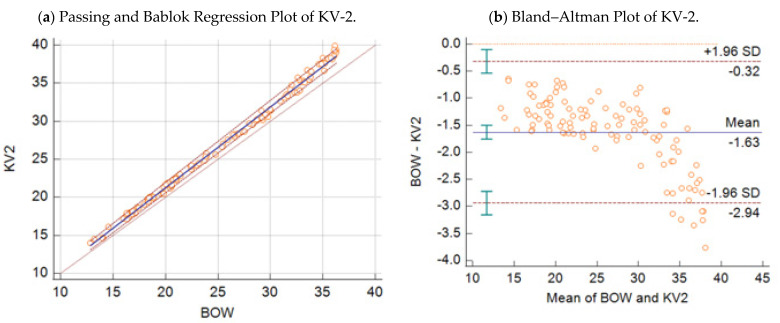
(**a**) Passing–Bablok regression plot: comparison of Ct values obtained with the wild-type KV-2 strain assay vs. CaPV-specific real-time PCR Bowden assay, with 95% CI in 109 samples. (**b**) Bland–Altman plot: comparison of Ct values obtained with the wild-type KV-2 strain assay vs. Capripox-specific real-time PCR Bowden assay, with 95% CI in 109 samples.

**Figure 2 microorganisms-09-01234-f002:**
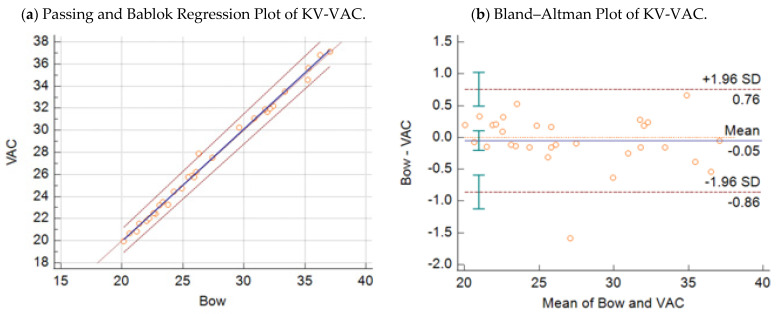
(**a**) Passing–Bablok regression plot: comparison of Ct values obtained with the KV-VAC Scheme 95. CI in 30 samples. (**b**) Bland–Altman plot: comparison of Ct values obtained with the KV-VAC strain assay vs. Capripox-specific real-time PCR Bowden assay, with 95% CI in 30 samples.

**Figure 3 microorganisms-09-01234-f003:**
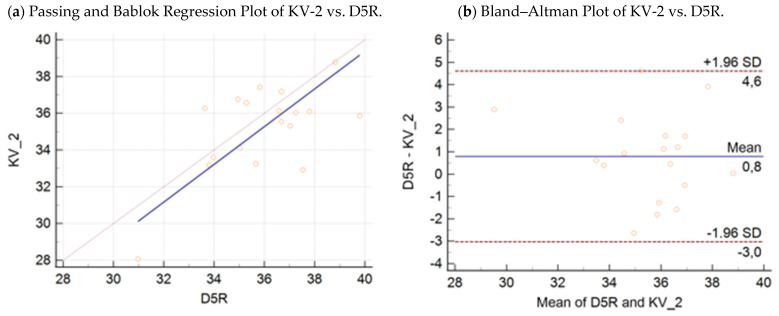
(**a**) Passing–Bablok regression plot: comparison of Ct values obtained with the wild-type KV-2 strain assay vs. Capripox-specific real-time PCR D5R assay, with 95% CI in 18 samples. (**b**) Bland–Altman plot: comparison of Ct values obtained with the wild-type KV-2 strain assay vs. Capripox-specific real-time PCR D5R assay, with 95% CI in 18 samples.

**Figure 4 microorganisms-09-01234-f004:**
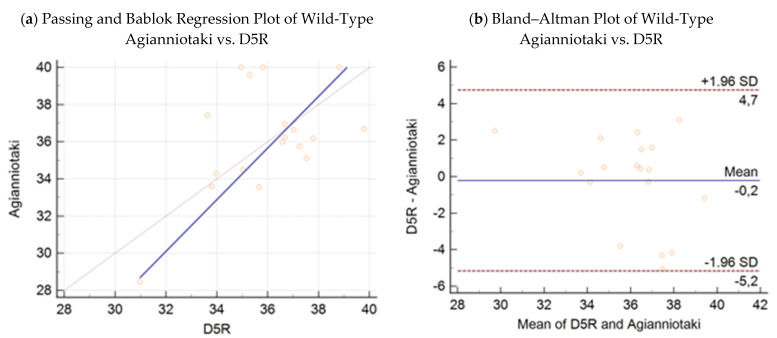
(**a**) Passing–Bablok regression plot: comparison of Ct values obtained with the wild-type Agianniotaki strain assay vs. Capripox-specific real-time PCR D5R assay, with 95% CI in 18 samples. (**b**) Bland–Altman plot: comparison of Ct values obtained with the wild-type Agianniotaki strain assay vs. Capripox-specific real-time PCR D5R assay, with 95% CI in 18 samples.

**Figure 5 microorganisms-09-01234-f005:**
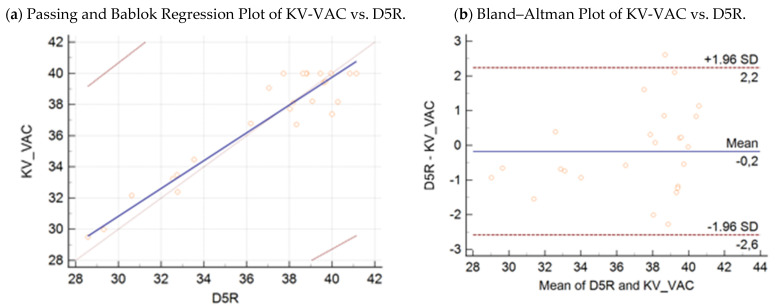
(**a**) Passing–Bablok regression plot: comparison of Ct values obtained with the KV-VAC strain assay vs. Capripox-specific real-time PCR D5R assay, with 95% CI in 25 samples. (**b**) Bland–Altman plot: comparison of Ct values obtained with the KV-VAC strain assay vs. Capripox-specific real-time PCR D5R assay, with 95% CI in 25 samples.

**Figure 6 microorganisms-09-01234-f006:**
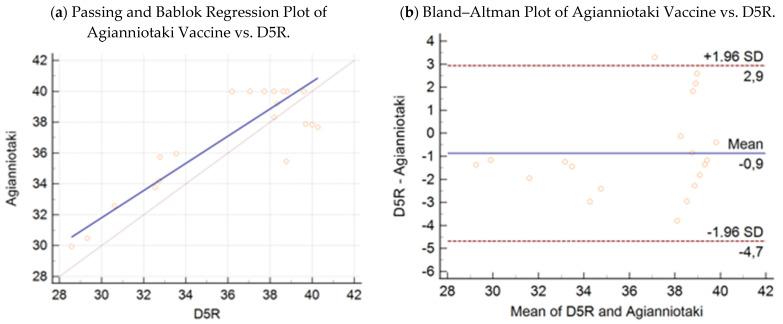
(**a**) Passing–Bablok regression plot: comparison of Ct values obtained with the Agianniotaki vaccine strain as-say vs capripox specific real-time PCR assay D5R, with 95% CI in 20 samples. (**b**) Bland–Altman plot: comparison of Ct values obtained with the Agianniotaki vaccine strain assay vs capripox specific real-time PCR assay D5R, with 95% CI in 20 samples.

**Table 1 microorganisms-09-01234-t001:** Primers and probe for the KV-2 assay specific for wild-type strains of LSD virus.

Oligo	Sequence (5′–3′)
KV2_Fmod	TGGGAYGATAACAACGTTTATG
KV2_Rmod	ACATTGTCATCYGGTAATGTA
LSD_KV2_Pro_field	VIC-TTACCACCTAATGATAGTGTTTATGATTTACC-BHQ1

**Table 2 microorganisms-09-01234-t002:** Primers and probe for the KV-VAC assay specific for Neethling-based vaccine strains of LSD virus.

Oligo	Sequence (5′–3′)
LSDV_vacc_5670f	TGCTTGTTTCCATTCTCCACT
LSDV_vacc_5829r	AAAAATGGGCGCAGTAGTATTT
LSDV_vacc_5726_Pro	FAM-CGCTGACATCGTTAGTCCACTCG-BHQ1

**Table 3 microorganisms-09-01234-t003:** Number and type of clinical samples used in assay validation.

**Samples Collected during LSD Outbreak in 2016**	
1. Skin biopsy	94
2. Blood	31
3. Nasal swabs	17
4. Milk	2
5. Flies	7
6. Environmental swabs	12
**Samples Collected during Prior Outbreak (2014) and after the Outbreak (2017)**	
7. DNA of internal organs originating from cattle taken in 2014	40
8. Blood of animals vaccinated in 2017	40
Total	243

**Table 4 microorganisms-09-01234-t004:** Repeatability of the KV-2 assay.

	Run 1	Run 2	Run 3
Mean	35.73125	35.17375	35.43875
SDEV	0.410311	0.292913	0.285329
CV	1.148326	0.832761	0.805132
Inter-run mean	35.44792		
Inter-run SDEV	0.278863		
Inter-run CV	0.79		

**Table 5 microorganisms-09-01234-t005:** Repeatability of the KV-VAC assay.

	Run 1	Run 2	Run 3
Mean	35.55375	35.285	35.19625
SDEV	0.227843	0.314234	0.352134
CV	0.640841	0.890559	1.000487
Inter-run mean	35.345		
Inter-run SDEV	0.167818		
Inter-run CV	0.47		

**Table 6 microorganisms-09-01234-t006:** Diagnostic sensitivity and diagnostic specificity for the KV-2 and KV-VAC assays vs. Bowden assay.

	Diagnostic Sensitivity and Diagnostic Specificity for KV-2 Assay vs. Bowden Assay		Diagnostic Sensitivity and Diagnostic Specificity for KV-VAC Assay vs. Bowden Assay	
Statistic	**Value**	**95% CI**	**Value**	**95% CI**
Sensitivity	97.46%	92.75–99.47%	100.00%	89.11–100.00%
Specificity	100.00%	92.60–100.00%	100.00%	97.22–100.00%
Positive likelihood ratio
Negative likelihood ratio	0.03	0.01–0.08	0
Disease prevalence	95.00%	95.00%
Positive predictive value	100.00%	100.00%
Negative predictive value	67.43%	40.38–86.35%	100.00%
Accuracy	97.58%	93.94–99.34%	100.00%	97.76–100.00%

**Table 7 microorganisms-09-01234-t007:** Additional testing on vaccinated animals and negative animals.

Samples	Number of Samples	Bowden Positive	Bowden Negative	KV-2 Positive	KV-2 Negative	KV-VAC Positive	KV-VAC Negative
DNA from true negative cattle (samples from 2014)	40	0	40	0	40	0	40
Blood from vaccinated cattle 2017	40	12	28	0	40	12	28

All samples from cattle that were negative for LSDV (*n* = 40) tested negative when using the Bowden, KV-2, and KV-VAC assays. From samples that originated from vaccinated animals (*n* = 40), 12 were positive when using the Bowden and KV-VAC assays, and none were positive when using the KV-2 assay.

**Table 8 microorganisms-09-01234-t008:** Repeatability and reproducibility of the KV-2 assay.

KV-2 (WT)	Day 1	Day 2	Day 3	Day 4	Day 5
Mean	35.5	34.95	34.68	34.45	34.43
Std. Deviation	0.2449	1.237	0.6397	0.4509	0.5737
Coefficient of variation	0.69%	3.54%	1.85%	1.31%	1.67%
Inter-run mean	34.8				
Inter-run SD	0.7525				
Inter-run CV	2.16%				

**Table 9 microorganisms-09-01234-t009:** Repeatability and reproducibility of the KV-VAC assay.

KV-VAC (VAC)	Day 1	Day 2	Day 3	Day 4	Day 5
Mean	32.5	31.1	32.05	31.7	32.13
Std. Deviation	0.383	0.469	0.5686	0.2708	0.5315
Coefficient of variation	1.18%	1.51%	1.77%	0.85%	1.65%
Inter-run mean	31.9				
Inter-run SD	0.632				
Inter-run CV	1.98%				

**Table 10 microorganisms-09-01234-t010:** Diagnostic sensitivity and diagnostic specificity for the KV-2 and KV-VAC assays vs. D5R assay.

	Diagnostic Sensitivity and Diagnostic Specificity for KV-2 Assay vs. D5R Assay		Diagnostic Sensitivity and Diagnostic Specificity for KV-VAC Assay vs. D5R Assay	
Statistic	**Value**	**95% CI**	**Value**	**95% CI**
Sensitivity	100.00%	81.47–100.00%	85.71%	69.74–95.19%
Specificity	100.00%	88.43–100.00%	100.00%	81.47–100.00%
Positive likelihood ratio
Negative likelihood ratio	0.00	0.14	0.06–0.32
Disease prevalence	95.00%	95.00%
Positive predictive value	100.00%	100.00%
Negative predictive value	100.00%	26.92%	14.06–45.34%
Accuracy	100.00%	92.60–100.00%	86.43%	74.22–94.28%

**Table 11 microorganisms-09-01234-t011:** Diagnostic sensitivity and diagnostic specificity for the Agianniotaki assay vs. D5R assay.

	Diagnostic Sensitivity and Diagnostic Specificity for Wild-Type Agianniotaki Assay vs. D5R Assay		Diagnostic Sensitivity and Diagnostic Specificity for Agianniotaki Vaccine Assay vs. D5R Assay	
Statistic	**Value**	**95% CI**	**Value**	**95% CI**
Sensitivity	100.00%	81.47–100.00%	75.00%	58.80–87.31%
Specificity	100.00%	88.43–100.00%	100.00%	81.47–100.00%
Positive likelihood ratio
Negative likelihood ratio	0.00	0.25	0.15–0.43
Disease prevalence	95.00%	95.00%
Positive predictive value	100.00%	100.00%
Negative predictive value	100.00%	17.39%	10.96–26.48%
Accuracy	100.00%	92.60–100.00%	76.25%	63.26–86.44%

## Data Availability

All the data that is not present in the paper can be obtained upon request to the authors.

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
