# Peer review of "Validation of TaqMan-Based Assays for Specific Detection and Differentiation of Wild-Type and Neethling Vaccine Strains of LSDV"

_microorganisms, 2021, doi:10.3390/microorganisms9061234_

Round 1

Reviewer 1 Report

I agree with authors that the research results presented in the paper are of a great importance for improving the efectiveness of LSD control with the use of vaccination. The authors have professionally presented the procedures and validation results of KV-DIVA qPCR protocol for rapid, sensitive and specific detection and differentiation of wild strains of LSDV from Neethling based vaccine strains of LSDV. The authors clearly presented the design of KV-2 and KV-VAC assays refering also to the previous publications. They also applied appropriate validation study procedure, including DNA extraction, samples for analytical sensitivity, specificity, efficiency and repeatability. It should be emphasized, they collaborated with several research centers. The samples for the sensitivity and specificity tests were appropriately selected. Also, the comparative analysis samples were correctly selected. The Real-time PCR procedurÄ™ is also clearly presented. It should be emphasized that the authors presents an extensive validation study  - 90 different Capripox virus strains and 291 samples originated from naturally infected animals, vaccinated animals, insects and environmental samples. The article should be published after editorial and probably liguistic correction.

Author Response

Response to reviewer 1, for revision of manuscript microorganisms-1184588

Dear reviewer,

Authors would like to thank the reviewer for careful and thorough reading of this manuscript. We appreciate the time and effort that you dedicated to providing feedback on our manuscript. The comments are encouraging and the reviewer appear to share our judgement that this study and its results are important for all scientists that are dealing with Lumpy skin disease.

Best regards,

Authors

Reviewer 2 Report

Revision of manuscript microorganisms-1184588

Dear Authors,

Your manuscript entitled “Validation of Taqman based assays for specific detection and differentiation of wild type and Neethling vaccine strains of LSDV” describes a DIVA assay to distinguish Neethling vaccine strain from field strains of LSDV. Although the validation of new diagnostic test is of sure interest, in my opinion, You did not well and sufficiently stress the importance and utility of your study: the proposed protocol can not be used as first line screening test and it has the same performance of the compared protocol (Agianniotaki), as reported by the same Authors in the discussion; so, it is not clear the useful of the developed protocol, probably some more appropriate explanations must provided and this aspect must be emphasized.

Results section is very confusing, too much long, dispersive and there are too many sections and sub-sections.

There are too many tables in the manuscript and in particular in Results section; tables should help readers to visualize the most relevant information of the study (material used or obtained results); all these tables, each one with a different (bad) style confuse the reader and probably the same Authors (sorry for this comment) considering that table’s numbers were at random.

Along all manuscript Authors stressed “where the experiments were carried out”; this is very confusing and a (good) reason of this was provided only at the end of the manuscript, line 472; probably this concepts should be introduced in introduction section and highlighted in the aim.

Below you can find detailed comments:

  • Introduction:
    • Line 43: here is the first time Authors use the virus acronyms SPPV and GTPV, please add full name here or abbreviation above.
  • Material and Methods:
    • Table 4 and 5: please, modify these table to be more elegant, understandable and suitable for the readers. Table 4: modify the caption and add a firs line with description of the columns content; if it is possible add information for each strains (year, animals and please of isolation for examples; a column reporting if are filed strains, reference strains, vaccine strains could be useful to immediately visualize the origin/characteristic of the strains), and for this particular reasons, consider to move this table to Supplementary material. In table 5 some more information are olase appreciated; “Samples tested by Sciensano are marked with *” in clude this a s legend and not in the title; I suggest to writhe “List of other microorganisms used for analytical specificity”; also for this table evaluate the possibility to move it in Supplementary material.
    • Page 5: it is table 6 not 3; please modify stile and title of the table.
    • Lines 168-170: I agree with Authors: in the experiment development phase it is important to test the 2 assays separately. Hoover, Did the Authors, in a second step, test the assay as duplex? I think it is important to verify all work well, considering this is the standard protocol that should be used.
  • Results:
    • Page 7: numbers of Tables are wrong. A better title for the table must be provided.
    • Page 7, Table 6: please, evaluate the possibility to move the table in Supplementary material and providing a descriptive paragraph summarizing the main obtained results of this part of the experiment (or a short table summarizing the results, for example grouping the samples for type). Furthermore, I did not understand some results, for example line 29: the sample was negative in all 3 assays, but you concluded (qPCR conclusion column) it is positive for Would type strain; please give an explanation for this. The same for line 34, 49, 54, 75.
    • Page 12 “3.1.7. Overview and summary of comparative testing”: is this a separate sub-paragraph? It has the number, but it has not the title. Furthermore, it is essential to describe the obtained results (a short description is enough), it is not possible to write a paragraph saying that results are present in the table.
    • Page 12 Table 7: title is generic and not explicative of the table content.
    • Page 12 “3.1.8 Diagnostic sensitivity and diagnostic specificity”: same as above.
    • Page 12 Table 8: same as above; furthermore, the stile of the table must be improved to make it easier to read and understand.
    • Page 13 Table 9: same as above.
    • Page 16 “3.11 An additional exclusivity testing”: it is 3.1.11? the title of this sub-paragraph is missing.
    • Page 17 “3.2.2. Comparative testing”: same as above.
    • Table 11, 12 and 13 have no title.
    • Please, consider the possibility to move Table 13 in Supplementary material.
    • Page 18 section 3.2.3 and Table 14: same as above.
    • Section 3.2.4: a brief summary of obtained results must be given. Table 15 and 15 6 titles are not acceptable.
    • Dear Authors, it is not possible that each table has a different style.
    • Table 17 and 18, same as above.
    •  
  • Discussion
    • Line 403-417: this introductive part to the discussion is interesting, but it is too much long; Authors are invited to focus on obtained results.

I sincerely hope that these suggestions will enhance this manuscript. However, if I have made any errors or misinterpretations, I apologize in advance.

Sincerely

The Reviewer

Round 2

Reviewer 2 Report

Revision of manuscript microorganisms-1184588-V2

Dear Authors,

You presented a different manuscript this time: same experiments and same data, but better presented.

Now the manuscript is easy to read and to understand; this new stile valorize your work. The aim is satisfactory and well explain the objective of the study.

However, I am sorry to bother you, but I have some more questions. I can not see the Supplementary material; I have access only to Table S1, I don’t know if some errors occurred or if I was wrong in material download.

If possible, I would be glad to see a clear version of the manuscript for a final check without correction.

Finally, evaluate the possibility to adda some information about the problems encountered in duplex protocol. The explanation you provided to me is reasonable and it doesn’t decrease the value of your work. If I read your manuscript as it is now and I decide to use your protocol as a duplex, probably, I will have problems. So it is reasonable to warm the reader about this, or simply delete the part where you say that the assay was developed to be used as a duplex.

Below you can find some very minor comments:

  • Material and Methods:
    • Table 3: evaluate the possibility to change title in something similar to “Number and type of clinical samples employed in assay validation”.
    •  

I sincerely hope that these suggestions will enhance this manuscript. However, if I have made any errors or misinterpretations, I apologize in advance.

Sincerely

The Reviewer

Author Response

Dear reviewers,

Please see our response in the attachment.

Best regards,

Authors

Round 3

Reviewer 2 Report

No more comments.

Good work!